# High Density Display of an Anti-Angiogenic Peptide on Micelle Surfaces Enhances Their Inhibition of αvβ3 Integrin-Mediated Neovascularization In Vitro

**DOI:** 10.3390/nano10030581

**Published:** 2020-03-22

**Authors:** Rajini Nagaraj, Trevor Stack, Sijia Yi, Benjamin Mathew, Kenneth R Shull, Evan A Scott, Mathew T Mathew, Divya Rani Bijukumar

**Affiliations:** 11601 Parkveiw Ave, Regenerative Medicine and Disability Research Lab, Department of Biomedical Sciences, University of Illinois College of Medicine at Rockford, Rockford, IL 61107, USA; rnagar5@uic.edu (R.N.); mtmathew@uic.edu (M.T.M.); 2Department of Biomedical Sciences, Northwestern University, Evanston, IL 60208, USA; trevorstack2019@u.northwestern.edu (T.S.); sijia.yi@northwestern.edu (S.Y.); k-shull@northwestern.edu (K.R.S.); evan.scott@northwestern.edu (E.A.S.); 3Department of Ophthalmology and Visual Sciences, University of Illinois Chicago, Chicago, IL 60612, USA; bmathew3@uic.edu

**Keywords:** Micelles, PEG-b-PPS, anti-angiogenic, integrin, VEGF

## Abstract

Diabetic retinopathy (DR), Retinopathy of Pre-maturity (ROP), and Age-related Macular Degeneration (AMD) are multifactorial manifestations associated with abnormal growth of blood vessels in the retina. These three diseases account for 5% of the total blindness and vision impairment in the US alone. The current treatment options involve heavily invasive techniques such as frequent intravitreal administration of anti-VEGF (vascular endothelial growth factor) antibodies, which pose serious risks of endophthalmitis, retinal detachment and a multitude of adverse effects stemming from the diverse physiological processes that involve VEGF. To overcome these limitations, this current study utilizes a micellar delivery vehicle (MC) decorated with an anti-angiogenic peptide (aANGP) that inhibits αvβ3 mediated neovascularization using primary endothelial cells (HUVEC). Stable incorporation of the peptide into the micelles (aANGP-MCs) for high valency surface display was achieved with a lipidated peptide construct. After 24 h of treatment, aANGP-MCs showed significantly higher inhibition of proliferation and migration compared to free from aANGP peptide. A tube formation assay clearly demonstrated a dose-dependent angiogenic inhibitory effect of aANGP-MCs with a maximum inhibition at 4 μg/mL, a 1000-fold lower concentration than that required for free from aANGP to display a biological effect. These results demonstrate valency-dependent enhancement in the therapeutic efficacy of a bioactive peptide following conjugation to nanoparticle surfaces and present a possible treatment alternative to anti-VEGF antibody therapy with decreased side effects and more versatile options for controlled delivery.

## 1. Introduction

Diabetic retinopathy (DR), Retinopathy of prematurity (ROP), and Age-related macular degeneration (AMD) are the most common diseases affecting the posterior segment of the eye [1,2,3]. By the end of 2030, approximately 350 million patients will be afflicted by these conditions. Neovascularization, the elevated growth of blood vessels [4], is common to the pathophysiology for DR, ROP, and wet AMD. In healthy eyes, there exists a sophisticated balance between pro-angiogenic and anti-angiogenic factors that regulate and strictly controls the formation of new blood vessels. In the disease state, prolonged abnormal growth of blood vessels in the retina leads to retinal ischemia, hypoxia and subsequent imbalance between pro-angiogenic and anti-angiogenic factors. Pigment epithelium-derived factor (PEDF) is an anti-angiogenic agent expressed by the retinal pigment epithelium cells that inhibit the formation of new blood vessels by blocking vascular endothelial growth factor (VEGF) [5]. The increased expression of hypoxia-inducible factor-1α (HIF-1α) during hypoxia leads to elevated expression of pro-angiogenic factors including VEGF, which is one of the most critical and prominent inducers of neovascularization [6,7,8,9,10]. Neovascularization leads to an increase in fluid accumulation and retinal hemorrhage in the posterior segment of the eye that can cause vision impairment. This condition can also result in retinal detachment and death of photoreceptors that can cause permanent loss of vision.

Anti-VEGF therapy is considered to be one of the most effective treatment options for DR and wet-AMD and is frequently used off label to treat ROP [11,12,13,14,15]. Current FDA approved anti-VEGF agents include bevacizumab, ranibizumab, and aflibercept, which are administered via monthly or bi-weekly intravitreal injections [14,16,17,18]. However, the treatment of posterior segment diseases involves more frequent drug administrations that can result in poor patient compliance [19,20,21,22,23]. In addition, these repeated injections are associated with increased chances of endophthalmitis, rhegmatogenous retinal detachment, hemorrhage, increase in intraocular pressure, ocular inflammation, blurred vision, and cataract [24]. High dosages of anti-VEGFs can cause systemic complications like decreased serum VEGF levels, myocardial infarction, CNS hemorrhage, and ocular complications such as tears in the retinal pigment epithelium, retinal detachment, and geographic atrophy [20]. In the case of ROP, anti-VEGF therapy decreases the development of neural and vascular structures and neurocognitive disorders. Furthermore, leaking of anti-VEGF from the vitreous to systemic circulation can also affect organ development in preterm infants [25,26,27]. New therapeutics and delivery systems are therefore needed to overcome these challenges associated with anti-VEGF therapy, and anti-integrin therapy has been recently investigated as an alternative [28,29,30].

Despite angiogenesis being partially regulated by specific integrin transmembrane glycoproteins, anti-integrin therapy remains underexplored as a treatment option for retinal pathologies [31,32]. Integrin α_v_β_3_ is highly expressed in angiogenic endothelial cells and tumors [31,33]. In ocular angiogenesis, integrin α_v_β_3_ has been found to be selectively expressed on the Choroid Neovascularisation (CNV) membrane [34,35,36,37,38]. Several studies revealed that integrin antagonists could be targets for ocular angiogenic disease as a next-generation treatment modality [39,40]. Anti-angiogenic peptides exhibiting a PDZ binding motif (*Ser-Asp-Val*) bind α_v_β_3_ integrin with high affinity and have previously demonstrated antiangiogenic effects on human endothelial cells. However, the therapeutic application of PDZ-containing peptides requires prohibitively high concentrations. We hypothesized that a nanoscale delivery vehicle that constitutively presented the peptide at high surface valency could enhance integrin clustering for therapeutically effective intracellular signaling at lower, more practical peptide concentrations.

In this study, self-assembled micellar nanocarriers composed of poly(ethylene glycol)-*b*-poly(propylene sulfide) (PEG-b-PPS) are validated as a platform to deliver an effective surface density of the anti-angiogenic peptide GHSDVHK. PEG-b-PPS self-assembles into a diverse range of nanocarrier morphologies that possess dense PEG coronas to inhibit nonspecific protein adsorption as well as hydrophobic PPS domains for retention of lipophilic payloads [41,42,43,44]. Recently, Stack et al. employed PEG-b-PPS micelles for targeted intracellular delivery of latrunculin A as a treatment for glaucoma [45]. Furthermore, Yi et al. demonstrated that lipidated peptides could be stably incorporated into PEG-b-PPS nanocarriers for in vivo applications and that the surface density of these peptides could be optimized using in vitro cell culture systems for enhanced binding to specific cell membrane receptors [46]. In this study, we present PEG-b-PPS micelles decorated with anti-angiogenic peptide (aANG-peptide) lipid constructs to modulate and study inhibition of α_v_β_3_ mediated neovascularization on endothelial cells.

## 2. Materials and Methods

### 2.1. Materials

The general materials utilized include PBS obtained from Gibco (Cat No. 10010023, Carlsbad, CA, USA), N-Methyl-2-pyrrolidone (NMP) purchased from Sigma Aldrich, St. Louis, MO, USA (Cat No. 328634) and Dichloromethane (DCM) from Sigma Aldrich, St. Louis, MO,, USA (Cat No. 270997). The other reagents required were Trifluoroacetic acid (TFA) (Sigma Aldrich, St. Louis, MO, USA – Cat No. T6508), acetonitrile (Sigma Aldrich, St. Louis, MO, USA Cat No. 75-05-8), diethyl ether (Sigma Aldrich, St. Louis, MO, USA – Cat No.309966), primary Human Umbilical Vein Endothelial Cell line (HUVECs) purchased from Sigma Aldrich (Cat No. S200-05N St. Louis, MO, USA). The other common reagents used for the experiments involve MTT reagent (3-(4,5-Dimethylthiazol-2-yl)-2,5-Diphenyltetrazolium Bromide) purchased from Thermo Fisher Scientific (Cat No. M6494, Waltham, MA, USA), Gibco DMEM (Dulbecco’s Modified Eagle Medium) (Cat No. 11965-084), Corning Fetal Bovine Serum (FBS) (Cat No. 35015-CV), basic Fibroblast Growth Factor (bFGF) purchased from Peprotech, Rocky Hill, NJ, USA (Cat No. 100-18B) and Apoptosis and necrosis was analyzed using a commercially available kit from BD Biosciences (Cat No. 556547, San Jose, CA, USA).

### 2.2. Synthesis of PEG-b-PPS Copolymer and Self-Assembly of Micelles

PEG-b-PPS block co-polymer was synthesized according to a previously established protocol [45]. Briefly, initiation of anionic ring-opening polymerization of PPS was facilitated by PEG thioacetate deprotected by sodium methoxide. The reaction was allowed to complete and subsequently, Alexa fluor 555 Maleimide was used to endcap the polymer. PEG-b-PPS micelles were formed using the co-solvent evaporation method according to a previously established protocol [47]. 20 mg of PEG-b-PPS co-polymer was dissolved in 1 mL of DCM and was added dropwise to 1 mL of sterile Phosphate Buffer Solution (PBS) that was stirred vigorously. After complete solvent evaporation (approximately 2 h), the resulting micelle solution was transferred to sterile 2 mL microcentrifuge tubes and stored at 4 °C for further analysis.

### 2.3. Synthesis of aANGP Peptide Constructs

Peptide modification was achieved through standard solid-phase peptide synthesis techniques. A lipid tail was linked to the N-terminus of the amino group of the aANGP. A crude Fmoc protected peptide with embedded resin was custom designed and purchased from Peptide 2.0. The PEG spacer Fmoc-N-amido-dPEG®_6_-acid (Quanta Biodesign Ltd, Plain city, OH, USA Cat No. 10063) was used to facilitate the conjugation to a Palmitoleic Acid (Cayman Chemical, Ann Arbor, MI, USA Cat No. 373-49-9) lipid tail. Deprotection and addition of residues were confirmed via the ninhydrin test at each intermediate step. Further to remove the resin from the modified aANGP, 10mL of solution containing 95% Trifluoroacetic acid (TFA) (Sigma Aldrich, St. Louis, MO, USA – Cat No. T6508), 2.5% Thiosalicyclic acid (TSA) (Sigma Aldrich, St. Louis, MO, USA – Cat No. T33200) and 2.5% ddH_2_O was added along with few drop drops of Hydrochloric acid (HCl) (Sigma Aldrich, St. Louis, MO, USA – Cat No. H1758) and allowed to shake for about 4 h. The removal of the resin was confirmed by Mass Spectrometry (MS) (Agilent Triple Quad 6490A, Chicago, IL, USA) and by a change in color of the solution to yellow. The aANGP was washed 5X with NMP to completely remove the cleaved resin. The solvent was evaporated using a rotary vaporizer, and the aANGP was precipitated using ice-cold diethyl ether (Sigma Aldrich, St. Louis, MO, USA – Cat No.309966). The resin cleavage and stability of the aANGP were further confirmed after precipitation by performing MS, and the cleaved aANGP was lyophilized and stored at 4 °C. To purify the cleaved aANGP, the lyophilized aANGP was reconstituted with 10 mL of 0.1% TFA and then injected to RP-HPLC (Waters 2545, Milford, MA USA) under isocratic mode with 2% TFA and 0.1% acetonitrile (Sigma, USA Cat No. 75-05-8) as mobile phase. The samples were collected at retention times of 42 and 45 min and the presence of the peptide was confirmed by MS. The fractions collected were lyophilized and stored at −20 °C. Liquid Chromatography-Mass Spectrometry (LC-MS) (Agilent Triple Quad 6490A, Milford, MA, USA) was performed to check for the purity and yield of the aANGP. 

### 2.4. Formulation of aANGP-MC

To prepare 1% molar ratio of aANGP to micellar delivery vehicles (MCs), 0.7 mg of the modified aANGP was added and rotated overnight in an end-to-end shaker at room temperature to allow the incorporation of the modified aANGP to the PEG-*b*-PPS micelles. The resulting aANGP MCs were purified by Sephadex LH 20 gravity chromatography column (Sigma Aldrich, St. Louis, MO, USA Cat No. GE17-0090-02) using PBS as a mobile phase. All the fractions were collected during purification and analyzed for the unconjugated aANGP. The resultant aANGP MCs obtained was subsequently concentrated using 10,000 MWCO Amicon® Ultra-4 Centrifugal Filter Units (EMD Millipore, Burlington, MA, USA, Cat No. C7719). To analyze the unconjugated aANGP in the fractions collected, the micro-BCA protein assay (Thermo Fisher, Rockford, IL, USA Cat No. 23235) was performed using BSA standards for obtaining a calibration curve.

### 2.5. Characterization of Developed Blank and aANGP-MCs

To determine the size of the micelles, Dynamic Light Scattering (DLS) (Malvern Instruments, Nano ZS90, Westborough, MA, USA) was performed which gives the size distribution profile of the particles in solution. The hydrodynamic radius was acquired by DLS based on the viscosity of 0.89 mPa and a refractive index of 1.5. The correlation function was measured at a scattering angle of 90° at T = 25 °C. To visualize the micelles, Cryogenic-Transmission Electron Microscopy (Cryo-TEM) was performed by using JEOL TEM-1230 Electron Microscope, Peabody, MA, USA.

### 2.6. Cell Culture

Human Umbilical Vein Endothelial Cells (HUVEC, Sigma, St. Louis, MO, USA) were used for analyzing the anti-angiogenic effect of the peptide. The cells were maintained in Endothelial Cell Growth Medium (Sigma, St. Louis, MO, USA Cat No. 211-500) with 1% Pen-Strep Antibiotic (Gibco, Carlsbad, CA, USA Cat No. 15140122) at 37 °C with 5% CO_2_. The seeding density of 1 × 10^6^ cells/mL was maintained in a T-75 flask. The cells at passage 3–6, were used for the experiment. Experimental plates were coated with denatured collagen coating solution (Sigma, St. Louis, MO, USA Cat No. 125-50). To determine the angiogenic properties of the HUVECs immunostaining was performed to detect two angiogenic markers vWF—von Willebrand Factor and PECAM-1—Platelet Endothelial Cell Adhesion Molecule-1. Primary antibodies mouse anti-human vWF (Sigma, St. Louis, MO, USA Cat No. F3520) and goat anti-human CD31/PECAM-1 (Novus Biologicals, Centennial, CO, USA Cat No. AF3628) was diluted to 1:1000 dilution in 1% BSA and incubated for 1 h. Secondary antibodies rabbit anti-mouse AlexaFluor 594 (Invitrogen, Carlsbad, CA, USA Cat No. A27027) and rabbit anti-goat AlexaFluor 488 (Invitrogen, Carlsbad, CA, USA Cat No. A-21222) of 1:5000 dilution in 1% BSA was added and incubated for 1 h in the dark and imaged using confocal microscopy (Make and model). 

### 2.7. Evaluation of Integrin αvβ3 Expression 

The target of the anti-angiogenic peptide in the current study is integrin αvβ3. Hence, the expression level of the integrin in HUVECs was evaluated using flow cytometry and confocal imaging. Since the expression of integrin αvβ3 requires stimulation of bFGF, the cells were incubated with 20 ng/mL of bFGF. Briefly, 1 × 10^6^ cells/well were seeded in a 6-well plate coated with denatured collagen and allowed to become confluent. The next day complete media was removed and starved with 0.5% FBS in DMEM with 20 ng/mL of bFGF and incubated for overnight (overnight starvation) and for 2 h (2 h starvation). The cells were then treated with Accutase to remove the adhered cells and centrifuged. The pellet was resuspended in 0.3% BSA in PBS. 5 μL of mouse anti-human CD51/CD6-FITC antibody (Invitrogen, Carlsbad, CA, USA Cat No. 11-0519-42) was added to the pellet and 5 μL mouse-IgG-FITC (Invitrogen, Carlsbad, CA, USA Cat No. 31501) was used as an isotype control. The cells that were not starved (cultured in normal endothelial growth media) was used as non-starved control. The samples were incubated in the dark for 30 min and then analyzed using FACS Caliber (BD Biosciences, San Jose, CA, USA) and imaged using confocal microscopy. 

### 2.8. Evaluation of Endothelial Cell Growth Inhibition 

Proliferation: In order to evaluate the effect of aANGP and aANGP MCs on the viability of the HUVECs, MTT assay was performed. Briefly, 5,000 cells/well were seeded in a 96-well plate coated with denatured collagen. On the next day, the cells were starved overnight with 0.5% FBS in DMEM with 20 ng/mL of bFGF. After starvation, the cells were treated with different concentrations of aANGP, aANGP MCs, and Blank MCs and incubated for 24 and 48 h. Untreated overnight starved cells were considered as control. After incubation MTT assay was performed by treating the cells with 10% MTT reagent (stock MTT reagent: 5 mg/mL) in media and incubated for 4 h. The MTT crystal formed was solubilized using MTT solubilizing solution and the absorbance was then measured at 570 nm and the results were analyzed. The percentage of viability/toxicity was calculated compared to control. The same experiment was conducted using non-starved + bFGF treated cells and starved +VEGF (20 ng/mL) treated cells.

Migration by Wound-Healing Assay: In order to evaluate the effect of aANGP and aANGP MCs on the migration of the HUVECs wound healing assay was performed. Briefly, 35,000 cells/well were seeded in a 24-well plate coated with denatured collagen. On the next day, the cells were starved overnight with 0.5% FBS in DMEM with 20 ng/mL of bFGF. After starvation, a scratch was made using 200 μL pipette tip and then the cells were treated with different concentrations of aANGP and aANGP MCs along with the blank MCs and incubated for 24 and 48 h. Untreated wells starved overnight with bFGF treatment were considered as control. Images of the scratch were taken at 0, 24, and 48 h and analyzed and quantified using T-Scratch Software (Adobe Systems v1.1, Inc. San Jose, CA, USA).

In-vitro Endothelial Cell Tube Formation Assay: HUVECs can form an intricate network in the presence of an extracellular matrix that contains collagen and other growth factors. The cells tend to form the network within 30 min of incubation, and complete tube formation will be achieved between 4 and 6 h [48]. In this experiment, the inhibition of angiogenesis in vitro was analyzed. HUVECs were starved overnight with 0.5% FBS in DMEM with 20 ng/mL of bFGF. The 96-well plate was coated with ECM Gel (Sigma, St. Louis, MO, USA Cat No. E1270) followed by incubation at 37 °C for 1 h in order to form a gel. 12,000 cells/well of was seeded on the coated plates then overnight starved HUVECs cells with different concentrations of aANGP and PEG-b-PPS-aANGP incubated at 37 °C for 4 h. The degree of tube formation was observed by inverted microscope and then analyzed by measuring the length of the tubes in each well using Angiogenesis Analyzer Plugin of Image J, and the data was represented compared to untreated control. 

Apoptosis Assay: In order to evaluate the mechanism by which the cells are undergoing death, apoptosis assay was performed by staining the cells with Annexin V-FITC and Propidium Iodide (PI). Briefly, 1 × 10^6^ cells/well were seeded in a 6-well plate coated with denatured collagen and allowed to become confluent. The next day the cells were starved overnight with 0.5% FBS in DMEM with 20 ng/mL of bFGF. The cells were then treated with different concentrations of aANGP and PEG-b-PPS-aANGP and incubated at 37 °C for 24 and 48 h. The cells were then stained with Annexin V-FITC and PI by following the procedure provided in the PE Annexin V Apoptosis Detection Kit I (BD Biosciences, USA Cat No. 559763). Assessment of the cells for apoptosis was performed by using Confocal microscopy, and FACS Caliber (BD Biosciences, San Jose, CA, USA), and the data were analyzed using FlowJo (FlowJo v10.6.2, LLC,, California, USA). 

### 2.9. Statistics

Data obtained was performed in triplicates in three independent experiments. Student’s *t*-test and one-way ANOVA with Tukey’s multiple comparison post hoc test (IBM SPSS) was used to compare the data for statistical significance among the experimental groups and *p* < 0.05 was considered statistically significant. All the values are presented as mean ± standard deviation (mean ± SD).

## 3. Results

### 3.1. Characterization of aANGP Micelles 

The successful synthesis of the lipidated aANGP peptide construct was verified using liquid chromatography-mass spectrometry (LCMS). Figure 1a (i) and (ii) shows the Total Ion Chromatograms (TIC) of fractions collected at 42 and 45 min retention times. The molecular weight of aANGP, PEG, Fmoc and palmitoleic acid is 779, 575, 222, and 254 Da, respectively. Further subtracting the weight of water molecules associated with PEG (18 Da) and palmitoleic acid (18 Da), the final molecular weight of the modified protein was theoretically calculated to be 1350 Da. This correlates well with the measured LC-MS peaks at both 42 and 45 min retention time that showed 1349.9074 and 1349.9066 Da, respectively (Figure 1b), which verified 99% purity. The final yield of the lipidated peptide construct was approximately 2 mg. 

The DLS spectrum obtained showed that the average particle size of the aANGP MCs and blank MCs are between 20–50 nm. In the case of PEG-b-PPS (blank), 69.2% of the micelles were in the range of 18–32 nm with an average size of 20 ± 5.5 nm (Figure 1c(i)) and 56.54% of aANGP MCs were in the range of 28–50 nm with an average size of 30 ± 6.7 nm (Figure 1c(ii)). The cryo-TEM images (Figure 1d (i)–(ii)) obtained confirmed the morphology of the micelles formed to be spherical with an average particle size of 20–30 nm for blank MCs and 20–50 nm for the aANGP MCs, which correlated with the DLS data obtained.

### 3.2. Activation of Integrin αvβ3 Expressions to Mimic the Hypoxic Condition

The aANGP peptide-mediated inhibition of endothelial cell growth and proliferation is primarily through αvβ3 integrins. The expression of endothelial cell markers vWF and PECAM-1 by HUVECs was detected both in the presence and absence of bFGF (Figure 2). Upregulation of αvβ3 integrins was obtained by overnight starvation of HUVECs followed by bFGF treatment, which was quantified by flow cytometry (Figure 3a (i)–(iii). A significant increase (*p* < 0.005) in the expression of αvβ3 integrins in overnight starved (37.6 ± 3.41 IU) and 2 h starved conditions (31.033 ± 0.3 IU) was observed compared to the control (6.15 ± 5.2 IU). In addition, when 2 h starved cells were treated with 2 mg/mL of aANGP (26.33 ± 0.37 IU), there was a significant reduction (*p* < 0.005) in the integrin αvβ3 expressions when compared to 2 h starvation.

The confocal images (Figure 3b (i)–(ii)) also showed a significant increase (*p* < 0.005) in the expression of integrin αvβ3 in overnight starved samples (6.94 ± 0.09 IU) when compared to 2 h starved (2.96 ± 0.03 IU) and non-starved conditions (1.86 ± 0.11 IU). However, not all cells expressed the integrin, and thus mixed populations of cells expressing integrin αvβ3 were obtained under overnight starved conditions. Since overnight starvation showed maximum expression when compared to 2 h starvation, the overnight starvation condition was used for all subsequent experiments. 

### 3.3. Inhibition of Endothelial Cell Proliferation and Growth by aANGP MCs

HUVECs starved and activated with bFGF were further treated with aANGP-MCs to assess the efficiency of growth inhibition compared to peptide alone. A dose-dependent decrease in the percentage of viable cells was observed after 24 and 48 h of treatment when compared to the control in both aANGP and aANGP MCs samples. Blank MCs did not show significant toxicity at both 24 and 48 h of treatment. There was a significant reduction (*p* < 0.000,05) in the viability at 24 and 48 h when compared to the control at all concentrations of aANGP (Figure 4c). aANGP-MCs showed significant reduction in the viability at 24 h for concentrations of 4 μg/mL (76.9 ± 0.25%) and 2 μg/mL (80.25 ± 0.19%) as shown in Figure 4d. However TC50 of aANGP peptide was obtained at 2 mg/mL (55.28 ± 0.3%) in comparison to 4 μg/mL (58.7 ± 0.67%) of aANGP-MCs, indicating that the efficiency of peptide interaction with cells in micellar form (Figure 4c,d). In addition, the results also showed that in the absence of starvation and bFGF (Figure 4a) and starvation + VEGF treatment (Figure 4b) the peptide did not evoke any reduction in cell growth, demonstrating the specificity of the peptide to integrin αvβ3 upregulated in the presence of bFGF.

### 3.4. Inhibition of Endothelial Cell Migration and Tube Formation

Migration is a crucial event in the process of angiogenesis. The effect of aANGP and aANGP-MCs on inhibition of migration was evaluated by wound healing assay after treatment with aANGP and PEG-b-PPS for 24 and 48 h. Figure 5a shows the migration of HUVECs when treated with different concentrations of aANGP and aANGP-MCs. The images showed inhibition in migration at 2000 μg/mL of aANGP compared to only 4 μg/mL of aANGP MCs at 24 and 48 h. Figure 5b,c shows the quantified data represented in a bar graph where the *x*-axis represents the different concentrations of aANGP in free form or present on aANGP-MCs. Significant (*p* < 0.005) inhibition in the migration of the HUVECs at 2000 μg/mL (18.73 ± 0.69%—24 h and 18.62 ± 0.39%—48 h) and 1000 μg/mL (17.89 ± 0.12%—24 h and 14.2 ± 0.93%—48 h) aANGP treated cells was observed when compared to the control (10.7 ± 0.93%— 24 h and 4.08 ± 0.87%—48 h) at 24 and 48 h. In the case of aANGP-MCs, the inhibition was evident (*p* < 0.005) at 4 μg/mL (17.32 ± 1.31%—24 h and 16.87 ± 0.86%—48 h) and 2 μg/mL (16.33 ± 1.91%—24 h and 12.78 ± 2.39%—48 h). 

Similar trends were shown by aANGP-MCs in inhibiting tube formation by endothelial cells. HUVECs treated with aANGP-MCs (Figure 6b) demonstrated higher efficiency in inhibiting tube formation compared to aANGP alone (Figure 6a) in a dose-dependent manner. However, the reduction in tube length was observed at a 1000-fold lower concentration in micellar form than free peptide alone. These results again confirmed the enhanced inhibitory effect of aANGP-MCs.

### 3.5. Evaluation of the Mechanism of Cell Death by Apoptosis Assay 

To determine the cellular death mechanism of HUVECs when treated with aANGP and aANGP-MCs, flow cytometry and confocal microscopy analysis of apoptosis were performed by staining with Annexin V-FITC/Propidium Iodide (PI). aANGP of concentration 2000, 500, and 10 μg/mL and aANGP-MCs of concentrations 4, 2, and 1 μg/mL were used for the study. From the flow cytograms obtained (Figure 7a), it is evident that the cells are undergoing apoptosis at all tested concentrations of aANGP and aANGP-MCs in a dose-dependent manner at 24 and 48 h (b and d). The dot plots where then quantified using Flow Jo which showed significant (*p* < 0.005) increases in the percentages of apoptotic cells at all concentrations of aANGP and aANGP MCs after 24 and 48 h when compared to the control (Figure 7b,c). The confocal images obtained also showed the cells to undergo apoptosis after treatment with aANGP (Figure 8a) and aANGP MCs (Figure 8b). Images acquired for 4 μg/mL of aANGP-MCs showed that the cells were undergoing both apoptosis and necrosis.

## 4. Discussion

Use of peptides to inhibit angiogenesis is a promising strategy since peptides offer lower immunogenicity, facile chemical modification and customization, lower costs of production, and conserved domains that play an important role as anti-angiogenic agents [49]. Integrin αvβ3, a transmembrane receptor expressed on endothelial cells stimulated by bFGF during pathological angiogenesis, plays an important role in neovascularization and can be an effective target to inhibit angiogenesis without causing systemic toxicity. Bang et al., have studied the role of the αvβ3-binding peptide sequence HSDVHK, which inhibits angiogenesis induced by bFGF stimulated HUVEC^18^. Major challenges associated with peptide therapy include in vivo instability, serum protein binding, short half-life, and the need for high concentration injections [50]. Conjugation of the peptide to the surfaces of polymeric nanocarriers can address many of these issues and improve the efficacy and versatility of peptide therapy [50,51,52,53]. Importantly, the small size of peptides allows their surface density to be precisely controlled, and increased peptide valency can be employed for improved target binding and selectivity [34,51,53,54,55,56,57,58]. While many cells share the same receptors, their expression level of these receptors can vary significantly. The resulting intracellular signaling and therapeutic effects of peptides targeting cell surface receptors are typically dependent on receptor clustering within the cell membrane [59]. Such clustering events require the surface density of the peptide on the nanoparticle to be specified for the cell surface concentration of the target receptor [60,61]. This current study involves the display of heptapeptide GHSDVHK on the surfaces of PEG-b-PPS micelles to increase the surface density of peptide, enhancing the receptor-mediated response at lower overall solution concentrations of the peptide. 

To facilitate the surface display of the anti-angiogenic heptapeptide, the peptide was modified by conjugation to a palmitoleic acid tail with the use of a PEG spacer. The incorporation of the lipidated aANGP into the hydrophobic membrane of PEG-b-PPS micelles was achieved at 95.16% loading efficiency and resulted in the exposure of aANGP on the outer micelle surfaces. This high valency surface display enhanced the interaction of aANGP with the active site of the target integrin αvβ3. The loaded peptide construct did not affect the structural characteristics of the micelles, which remained relatively unchanged from blank PEG-b-PPS micelles. 

Further, to evaluate the anti-angiogenic effect of aANGP-MCs and aANGP, primary HUVECs were used, which are a widely used model cell population for studying angiogenesis in vitro [62]. HUVECs were verified to express vWF and PECAM-1, which are two significant cell surface markers that play a vital role in angiogenesis. The results obtained clearly show the expression of these angiogenic markers both with and without the presence of growth factor bFGF. However, there was increased expression of vWF and PECAM-1 in the bFGF treated cells when compared to non-treated cells, indicating that bFGF induces elevated expression of angiogenic markers in endothelial cells as reported by Norgall et al. [63,64]. Further, increased expression of the αvβ3 integrins following overnight starvation with bFGF when compared to 2 h starvation with bFGF was observed. Similar results were reported by Bang et al. and Odrlijin et al. [65,66]. Interestingly, the images obtained showed that not all the cells were expressing the integrin and hence there was a mixed population of cells expressing integrin at different levels.

Anti-angiogenic therapy using cytotoxic angiogenesis inhibitors, such as peptides, have been employed to inhibit abnormal neovascularization by endothelial cells [35,67,68]. In this study, we targeted endothelial cells expressing αvβ3, which is a marker for hypoxia-mediated neovascularization. The cell viability data obtained showed that the concentration of aANGP required to cause a 50% decrease in the cell viability was significantly higher than the amount of aANGP required when presented at high valency in micellar form, indicating that the micelles are more efficient than the peptide treatment alone [69]. More importantly, there was no cytotoxicity induced by the peptide to the control endothelial cells (Figure 4a,b), suggesting a viable and non-toxic option for anti-angiogenic therapy. In addition, aANGP and aANGP-MCs both inhibited the tube formation of endothelial cells in vitro which is an important factor in angiogenesis, suggesting that aANGP is a potent anti-angiogenic peptide in bare and micelle conjugated form. The results obtained for migration, tube formation, and apoptosis clearly demonstrate the anti-angiogenic effect of aANGP in free form, and it is notable that the amount of aANGP required to elicit a functional effect was significantly less when aANGP was displayed in the micellar form. Compared to high valency decorated MCs, the solution concentration of free form peptide required to bind a similar density of integrins is significantly higher (Figure 8a) [70,71]. This is likely due to the high valency display of aANGP on polymeric micelles that can locally interact with cellular receptors to induce receptor clustering and intracellular signal propagation. Such interactions have been shown in many systems to induce adhesion of polymeric particles to the cell surface and subsequent membrane wrapping around the particles [72], which can promote receptor-mediated endocytosis [73]. (Figure 8b). These local binding events at the nano/bio interface of aANGP-MC and the HUVEC cell membrane may account for the order of magnitude lower dose of peptide required to cause inhibition of angiogenesis. 

## 5. Conclusions

In summary, we have presented a PEG-b-PPS micellar formulation that presents aANGP at a high surface density to enhance anti-angiogenic effects on endothelial cells by targeting their αvβ3 integrin receptors. The aANGP-MCs showed 1000-fold higher inhibition in proliferation, migration, and tube formation than aANGP in free form alone. This increased efficacy is consistent with prior literature demonstrating the enhancement of intracellular signaling and resulting cellular responses that can occur when ligands are displayed at higher valencies to induce sufficient receptor clustering. The aANGP-MC formulation may offer a novel treatment option to inhibit a wide range of angiogenesis-mediated eye pathologies and suggests that PEG-b-PPS nanoparticles may serve as efficient delivery systems for therapeutic peptides.

## Figures and Tables

**Figure 1 nanomaterials-10-00581-f001:**
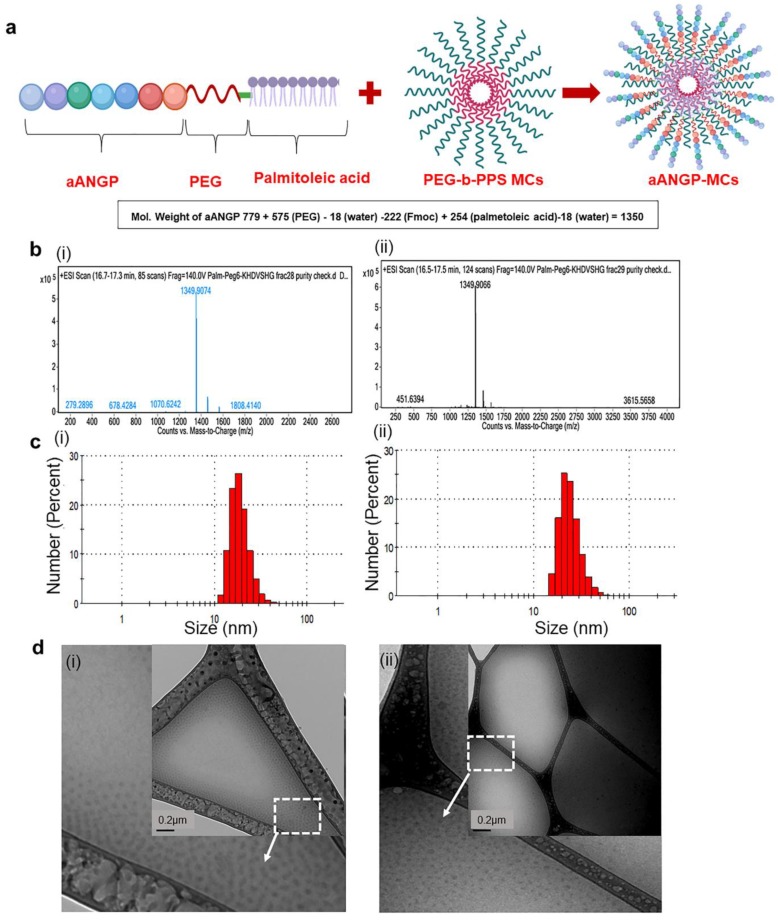
Design and characterization of aANGP-micellar delivery vehicle (MCs). (**a**) Schematic representation of the aANGP lipidated peptide construct and its insertion into poly (ethylene glycol)-*b*-poly (propylene sulfide) (PEG-b-PPS) MCs. (**b**) Total ion chromatograms of the fractions collected at retention times of 42 (i) and 45 (ii) min showing the peak obtained for the modified aANGP. (**c**) Dynamic Light Scattering (DLS) spectrum obtained for (i) PEG-b-PPS (blank) micelles and (ii) aANGP-MCs with the average particle size of 20–50 nm. (**d**) Cryogenic-Transmission Electron Microscopy (Cryo-TEM) images of (i) PEG-b-PPS (Blank) micelles and (ii) aANGP-MCs (inset) showed their respective morphology and size to both be spherical and 20–50 nm in diameter. Outer image is manually enlarged and cropped section of the white square part of the inset image.

**Figure 2 nanomaterials-10-00581-f002:**
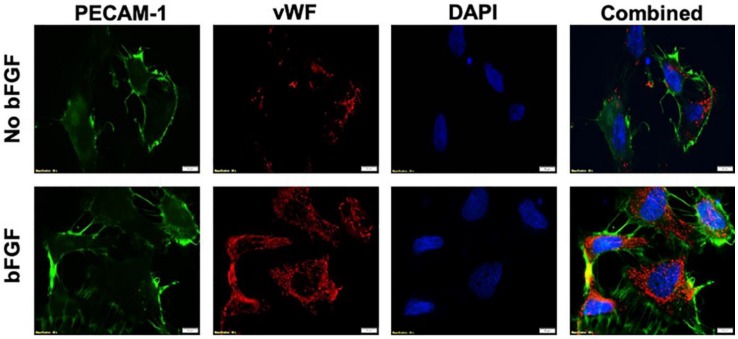
Immunostaining of angiogenic markers in Human Umbilical Vein Endothelial Cell line (HUVECs). Cells were stained with Platelet Endothelial Cell Adhesion Molecule-1 (PECAM-1) (green), vWF (red) and nucleus with DAPI (blue). The expression of PECAM-1 and von Willebrand Factor (VWF) is evident in HUVECs regardless of basic Fibroblast Growth Factor (bFGF) exposure.

**Figure 3 nanomaterials-10-00581-f003:**
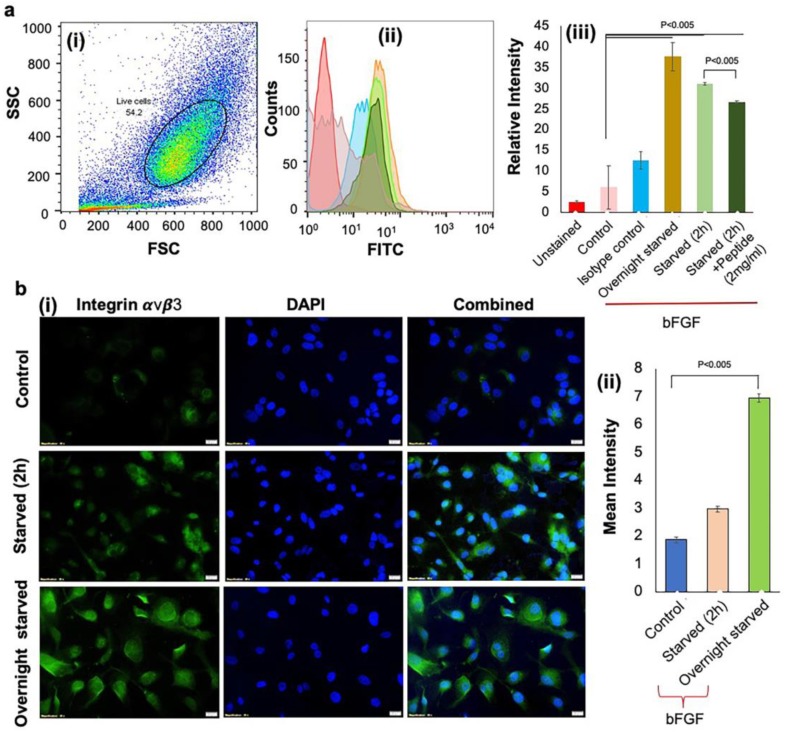
Expression of αvβ3 by HUVECs under different conditions (**a**) Flow cytometry analysis demonstrated significant expression of integrin αvβ3 in overnight starved cells. (i) Side scatter (SSC) vs. forward scatter (FSC) plot used to gate live cells and resulting (ii) histograms from which αvβ3 expression was (iii) quantified for different culture conditions. Colors in (ii) match the conditions shown in the x-axis of (iii). Values were expressed as mean ± SD, *n* = 3. *p* < 0.005 was considered significant. (**b**) Immunostaining was performed by (i) staining HUVECs with anti-human CD51/CD61 antibody to detect integrin αvβ3 (green). Nuclei were stained by DAPI (blue). Integrin αvβ3 expression was highest under overnight starved conditions when compared to control and 2 h starvation. (ii) Quantification of the images in (i) obtained using Image J. Value was expressed as mean ± SD, *n* = 3. *p* < 0.05 was considered significant.

**Figure 4 nanomaterials-10-00581-f004:**
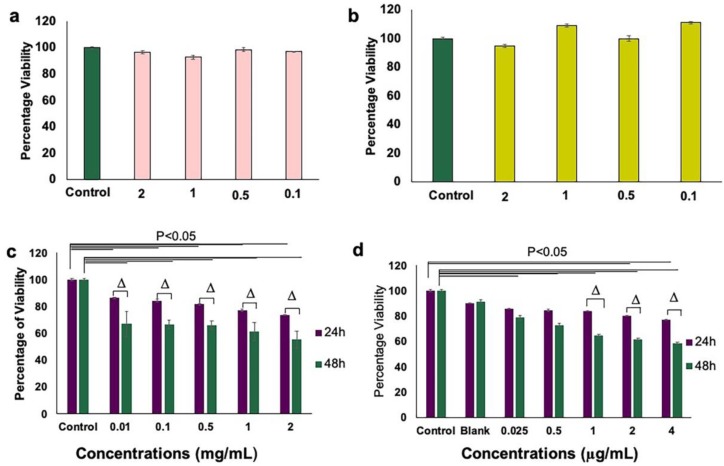
Comparison of HUVEC viability in the presence between aANGP-MCs and free form peptide as assessed by MTT assay. (**a**) Graph showing cell viability after treatment with aANGP without overnight starvation and (**b**) after treatment with aANGP with overnight starvation using 10 ng/mL of VEGF. (**c**) Viability results obtained after treatment with free form aANGP and (**d**) with aANGP-MCs after 24 and 48 h. Δ-Significance between the samples at 24 and 48 h. Values were expressed as mean ± SD, *n* = 3. Δ *p* < 0.05.

**Figure 5 nanomaterials-10-00581-f005:**
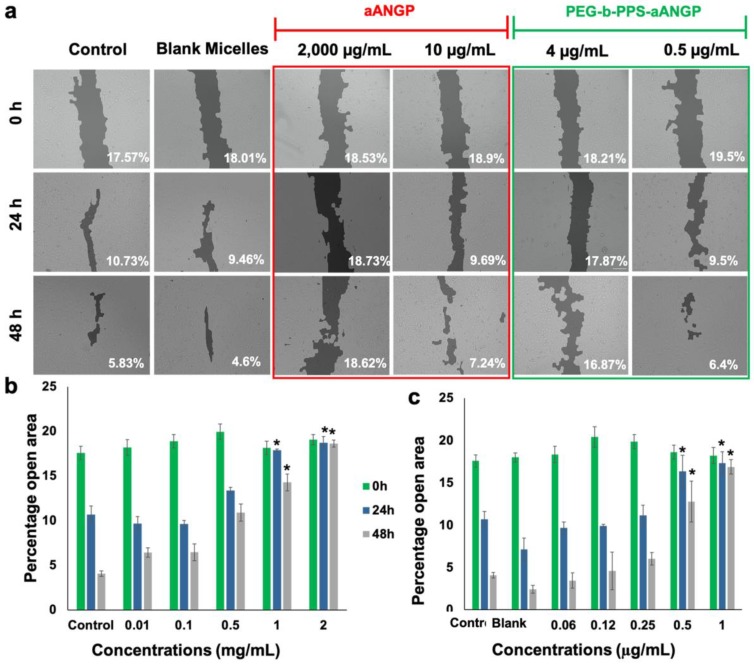
Effect of aANGP and aANGP-MCs on the migration of HUVECs. (**a**) HUVECs were treated with different concentrations of aANGP and aANGP-MCs for 24 and 48 h and imaged at 0, 24, and 48 h using an inverted microscope at 10× magnification. Bar graphs represent the quantification of the images obtained for (**b**) aANGP and (**c**) aANGP-MCs. Values were expressed as mean SD, *n* = 3. * *p* < 0.005.

**Figure 6 nanomaterials-10-00581-f006:**
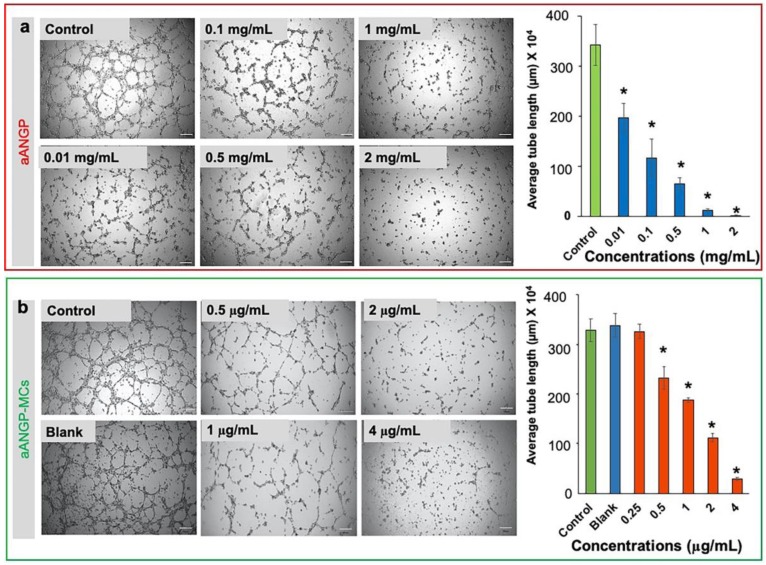
Effect of aANGP and aANGP-MCs on HUVEC tube formation: HUVECs were seeded on.plates coated with ECM gel, incubated for 4 h with different concentrations of aANGP and aANGP-MCs and then imaged using an inverted microscope at 4× magnification. Images obtained for treatment with different concentrations of (**a**) free form aANGP and (**b**) aANGP-MCs. Bar graph represents the quantification of the tubes formed obtained using Image J. Values were expressed as mean ± SD, *n* = 3. * *p* < 0.0005.

**Figure 7 nanomaterials-10-00581-f007:**
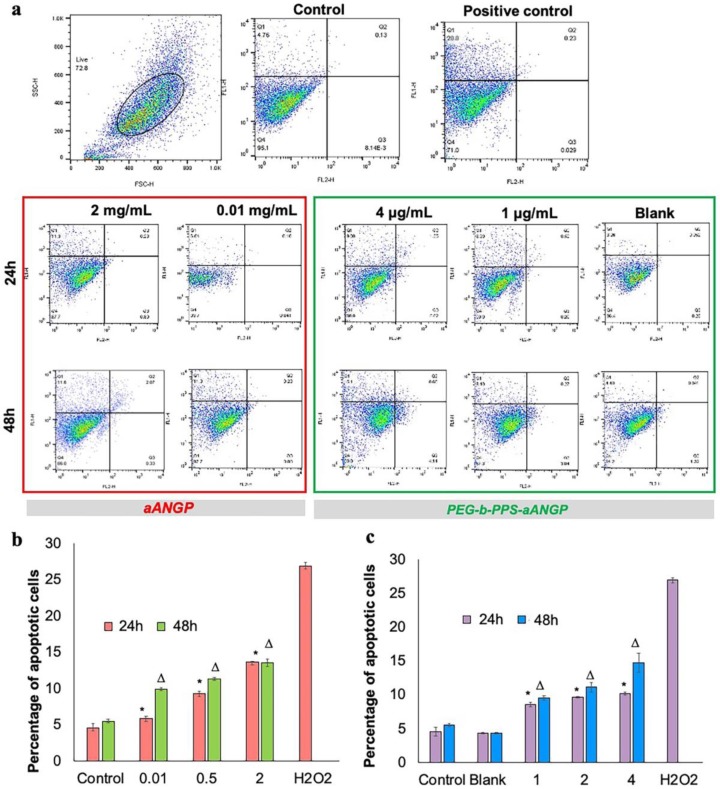
Apoptosis induced by aANGP and aANGP-MCs: HUVECs were treated with different concentrations of aANGP and aANGP-MCs for 24 and 48 h followed by Annexin V-FITC and PI staining. (**a**) Dot plots obtained after analyzing aANGP treated HUVECs using Flow cytometry at 24 and 48 h. Bar graphs represent the quantification of the dot plots obtained for (**b**) aANGP and (**c**) aANGP-MCs treated HUVECs were stained with Annexin V-FITC and PI and imaged using a fluorescent microscope at 20× magnification. Values were expressed as mean ± SD, *n* = 3. * (24h) and Δ (48h) * *p* < 0.05.

**Figure 8 nanomaterials-10-00581-f008:**
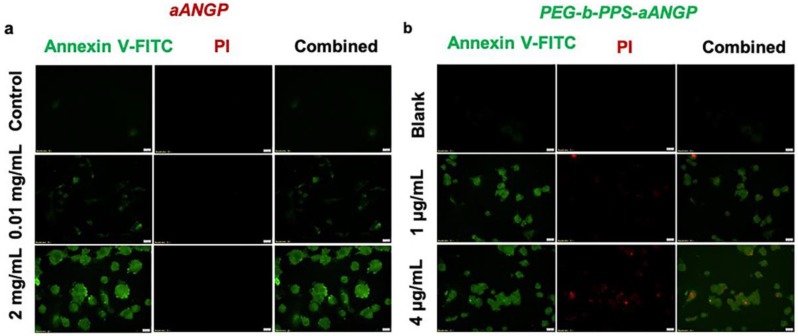
Confocal images of apoptosis induced by aANGP and aANGP-MCs: (**a**) Images obtained for aANGP and (**b**) aANGP-MCs. Green—Annexin V- FITC (apoptosis), red—PI (necrosis). Values were expressed as mean ± SD, *n* = 3. *p* < 0.005.

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
