# Peer review of "High Density Display of an Anti-Angiogenic Peptide on Micelle Surfaces Enhances Their Inhibition of αvβ3 Integrin-Mediated Neovascularization In Vitro"

_nanomaterials, 2020, doi:10.3390/nano10030581_

Round 1

Reviewer 1 Report

  1. I think the authors need to make their topic more specific as they are intended to apply for Diabetic retinopathy (DR), Retinopathy of Pre-maturity (ROP) and Age-related Macular Degeneration (AMD)
  2. The Introduction parts need more reference especially from the second paragraph onwards
  3. Figure 2d (TEM image) quality is so poor. It would  be better to show the micellar structure of the particles in high magnification TEM
  4. Authors need to explain the drug release mechanism 
  5. Is it possible to study the permeation effects using ex vivo pig retina?

Author Response

Reviewer #1:

Comment 1: I think the authors need to make their topic more specific as they are intended to apply for Diabetic retinopathy (DR), Retinopathy of Pre-maturity (ROP) and Age-related Macular Degeneration (AMD)

Answer: Thank you for the comment. You are correct in that applications of the developed micelles will include treatments for DR, AMD and ROP. However, the current study is focusing on the development and optimization of micelles that could one day lead to therapeutic strategies for the above-mentioned diseases, and these diseases are not directly addressed in this work. Hence, we kindly believe that it is appropriate to list these diseases as possible applications, even though the manuscript focuses specifically on the nanoparticle delivery system.

Comment 2: The Introduction parts need more reference especially from the second paragraph onwards

Answer: Thank you. We have included more references in the introduction in the revised manuscript.

Comment 3: Figure 2d (TEM image) quality is so poor. It would be better to show the micellar structure of the particles in high magnification TEM

Answer: The TEM images shown in Fig 2 are in 200nm resolution, which is comparable with the literature. However, we agree with the reviewer that due to the size of the image panel it will be hard for visibility. We have included an enlarged version of this TEM image as well as others in the supplementary file.

Comment 4: Authors need to explain the drug release mechanism

Answer: Thank you for the comment. The current study is to study the role of surface density of peptide on micelles to increase the efficiency of anti-angiogenesis. So the micelles have a therapeutic effect only when they are intact, and no drug is actually released.

Comment 5: Is it possible to study the permeation effects using ex vivo pig retina?

Answer: Thank you for the comment and excellent suggestion. We agree with the reviewer that an ex-vivo pig retina would be a good choice to study the efficiency in a future study. In the current work we employ the CAM assay to initially verify our results, but we plan to continue our research with an in vivo diabetic retinopathy mouse model.

Reviewer 2 Report

The paper by Kagaraj R and colleagues reports the characterization of anti-avb3 peptide decored micelles as antiangiogenic in HUVEC stimulated with bFGF. The authors demonstrate the higher potency of micelle constructs respect to isolated peptides, suggesting their potential use in angiogenesis dependent ocular diseases.

The paper is interesting, but this referee has some concerns

  • Regarding the general design of the experiments, have the authors any evidence on VEGF induced angiogenesis? Indeed in the introduction (and discussion) they pose attention on hypoxia induced VEGF upregulation and anti-VEGF drugs pharmacological effect. The possibility are two: or the authors provide additional experimental data also on VEGF induced responses or otherwise they have to mitigate the importance of VEGF in retina disease producing literature reports describing other angiogenic factor involved in ocular disorders (as FGF) and probably responsible for anti-VEGF therapy resistance. See doi: 1186/s40942-017-0084-9 as an example.
  • The title is not indicative of the paper. The word “valency” seems not appropriate and I suggest to rephrase it avoiding the abbreviation and introducing the concept that the study has been performed in vitro.
  • In the abstract there is no mention to the cellular model used in the study. Please indicate the use of HUVEC.
  • In the description of the results and also in the discussion (line 469-470) it is reported that starvation is the inducer for integrin upregulation. Since starvation (meaning lack of nutrients) is followed by bFGF stimulation, the observed effect is due to the addition of the angiogenic factor and not starvation alone. Moreover, what is the control in figure 3? Please check and specify better the experimental condition and result interpretation.
  • Line 216 of MTT test. Please provide the real concentration of MTT which is not properly reported.
  • In Figure2, 3 5,6 and 8 please add and/or indicate the dimension of the scale bar in the figure legends.
  • In Figure 3 please add indication in the panels and in the legend where bFGF has been used as a stimulus. Pay attention to figure 4 legend where VEGF has been reported as a stimulus!
  • Section 3.3 is related to figure 4 but there is no mention in the results to the single panels of the figure. Please add this in the text. Line 397 please check what is (b&d).
  • The first pert of the discussion (line 419-431) is a repetition of introduction. Please remove it and stress what is the novelty of the paper.
  • An extensive check of English grammar is necessary throughout the manuscript.

Author Response

Comment 1:  Regarding the general design of the experiments, have the authors any evidence on VEGF induced angiogenesis? Indeed, in the introduction (and discussion) they pose attention on hypoxia induced VEGF upregulation and anti-VEGF drugs pharmacological effect. The possibility are two: or the authors provide additional experimental data also on VEGF induced responses or otherwise they have to mitigate the importance of VEGF in retina disease producing literature reports describing other angiogenic factor involved in ocular disorders (as FGF) and probably responsible for anti-VEGF therapy resistance. See doi: 1186/s40942-017-0084-9 as an example.

Answer: Thank you for the comment. The role of VEGF in retinal diseases have been previously studied in detail by several researchers and is an established phenomenon (see references below).  Hence in this study we have not included any analysis pertaining to this pathology, as it is not the focus of this manuscript. The authors also agree with the reviewer that there are multiple factors contributing to neovascularization such as PDGF, PIGF, FGF, HGF, angiopoietin etc. However, the development of anti-VEGF therapy (current treatment strategy) over other targets (studied previously) clearly demonstrate the critical influence of VEGF in ocular diseases (Example: Feeney et al, 2003 Role of vascular endothelial growth factor and placental growth factors during retinal vascular development and hyaloid regression).  
References: Following are few references showing the role of VEGF in angiogenesis in eye.

Kusuhara S, Fukushima Y, Fukuhara S, Jakt LM, Okada M, Shimizu Y, Hata M, Nishida K, Negi A, Hirashima M, Mochizuki N. Arhgef15 promotes retinal angiogenesis by mediating VEGF-induced Cdc42 activation and potentiating RhoJ inactivation in endothelial cells. PloS one. 2012;7(9)
Rattner A, Williams J, Nathans J. Roles of HIFs and VEGF in angiogenesis in the retina and brain. The Journal of clinical investigation. 2019 Aug 12;129(9).
Youngblood H, Robinson R, Sharma A, Sharma S. Proteomic Biomarkers of Retinal Inflammation in Diabetic Retinopathy. International journal of molecular sciences. 2019 Jan;20(19):4755.
Li Y, Busoy JM, Zaman BA, Tan QS, Tan GS, Barathi VA, Cheung N, Wei JJ, Hunziker W, Hong W, Wong TY. A novel model of persistent retinal neovascularization for the development of sustained anti-VEGF therapies. Experimental eye research. 2018 Sep 1;174:98-106.
Sun M, Wadehra M, Casero D, Lin MC, Aguirre B, Parikh S, Matynia A, Gordon L, Chu A. Epithelial Membrane Protein 2 (EMP2) Promotes VEGF-Induced Pathological Neovascularization in Murine Oxygen-Induced Retinopathy. Investigative Ophthalmology & Visual Science. 2020 Feb 7;61(2):3- 

Comment 2: The title is not indicative of the paper. The word “valency” seems not appropriate and I suggest to rephrase it avoiding the abbreviation and introducing the concept that the study has been performed in vitro

Answer: As requested by the reviewer, we have provided a new title that does not include the word “valency” and includes the words “in vitro”:

“High density display of anti-angiogenic peptides on micelle surfaces enhances their inhibition of αvβ3 integrin-mediated neovascularization in vitro.”

Comment 3: In the abstract there is no mention to the cellular model used in the study. Please indicate the use of HUVEC.

Answer: Authors thank the reviewer for pointing out the missing information. We now included the model cells used in the study in the abstract.

Text change: “To overcome these limitations, this current study utilizes a micellar delivery vehicle (MC) decorated with an anti-angiogenic peptide (aANGP) that inhibits αvβ3 mediated neovascularization using primary endothelial cells (HUVEC).”

Comment 4: In the description of the results and also in the discussion (line 469-470) it is reported that starvation is the inducer for integrin upregulation. Since starvation (meaning lack of nutrients) is followed by bFGF stimulation, the observed effect is due to the addition of the angiogenic factor and not starvation alone. Moreover, what is the control in figure 3? Please check and specify better the experimental condition and result interpretation.

Answer: We apologize for the confusion. We agree with the reviewer that the upregulation of integrin is not only because of starvation but due to the presence of bFGF treatment. We stated this in the results section as “Upregulation of avb3 integrins was obtained by overnight starvation of HUVECs followed by bFGF treatment, which was quantified by flow cytometry (Fig. 3a).” However, this information was not very clear in the discussion part as well as figure legend. So we have modified our manuscript by adding those valuable points. The control of Fig 3 is the normal cell (non-starved) culture condition, i.e. cells cultured in endothelial growth medium that were already supplemented with optimum growth factors including bFGF (according to manufacturer’s instruction). A clear description of this control is now included in the revised manuscript.

Text Change: The cells that were not starved (cultured in normal endothelial growth media) were used as non-starved control.

Further, increased expression of the avb3 integrins following overnight starvation with bFGF when compared to 2 h starvation with bFGF treatment was observed.

Figure Legend change: “Integrin avb3 expression was highest under overnight starved conditions when compared to control and 2 h starvation followed by bFGF treatment. The non-starved cells without bFGF treatment (cells in endothelial cell growth medium) served as a control.”

Comment 5: Line 216 of MTT test. Please provide the real concentration of MTT which is not properly reported.

Answer: Thank you for the comment. We have used 10% MTT reagent (stock:5 mg/ml) in the complete media for assay. The information is updated in the methodology.

Text Change: . After incubation MTT assay was performed by treating the cells with 10% MTT reagent (stock MTT reagent: 5 mg/ml) in media and incubated for 4 h.

Comment 6: In Figure2, 3 5,6 and 8 please add and/or indicate the dimension of the scale bar in the figure legends.

Answer: We have updated the figure legends with scale bar dimensions. Thank you.

Text Change:

Figure 2: Immunostaining of angiogenic markers in HUVECs. A) Cells were stained with PECAM-1 (green), vWF (red) and nucleus with DAPI (blue). The expression of PECAM-1 and vWF is evident in HUVECs regardless of bFGF exposure. Scale bar: 10mm

Figure 3: Expression of avb3 by HUVECs under different conditions a) Flow cytometry analysis demonstrated significant expression of integrin avb3 in overnight starved cells. (i) Side scatter (SSC) vs. forward scatter (FSC) plot used to gate live cells and resulting (ii) histograms from which avb3 expression was (iii) quantified for different culture conditions. Colors in (ii) match the conditions shown in the x-axis of (iii). Values were expressed as mean ± SD, n=3. p<0.005 was considered significant. b) Immunostaining was performed by (i) staining HUVECs with anti-human CD51/CD61 antibody to detect integrin avb3 (green). Nuclei were stained by DAPI (blue). Integrin avb3 expression was highest under overnight starved conditions when compared to control and 2 h starvation followed by bFGF treatment. The non-starved cells without bFGF treatment (the cells in endothelial cell growth medium) was taken as control (ii) Quantification of the images in (i) obtained using Image J. Value was expressed as mean± SD, n=3. p<0.05 was considered significant. Scale bar: 20mm

Figure 5: Effect of aANGP and aANGP-MCs on the migration of HUVECs. (a) HUVECs were treated with different concentrations of aANGP and aANGP-MCs for 24 and 48 h and imaged at 0, 24 and 48 h using an inverted microscope at 10X magnification. Bar graphs represent the quantification of the images obtained for (b) aANGP and (c) aANGP-MCs. Values were expressed as mean ± SD, n=3. *p<0.005. The images were taken in 4x magnification.

Figure 6: Effect of aANGP and aANGP-MCs on HUVEC tube formation: HUVECs were seeded on plates coated with ECM gel, incubated for 4 h with different concentrations of aANGP and aANGP-MCs and then imaged using an inverted microscope at 4X magnification. Images obtained for treatment with different concentrations of (a) free form aANGP and (b) aANGP-MCs. Bar graph represents the quantification of the tubes formed obtained using Image J. Values were expressed as mean ± SD, n=3. p<0.0005. The images were taken in 4x magnification.

Figure 8: Confocal images of apoptosis induced by aANGP and aANGP-MCs: (a) Images obtained for aANGP and (b) aANGP-MCs. Green – Annexin V- FITC (apoptosis), red – PI (necrosis). Values were expressed as mean ± SD, n=3. p<0.005. Scale bar: 20mm

Comment 7: In Figure 3 please add indication in the panels and in the legend where bFGF has been used as a stimulus. Pay attention to figure 4 legend where VEGF has been reported as a stimulus!

Answer: The bFGF stimulus is now indicated in the figure legend and in figure 3. Thank you for pointing out the use of VEGF in the figure caption of 4b, but we actually did intend to write VEGF here as a stimulus.  Figure 4b shows a control group where starved cells stimulated with VEGF were also found to be affected by peptide treatment, which indicated the selectivity of the peptide towards avb3 integrins. To clarify, we also added an extra sentence to the methods section 2.8 to describe the VEGF experiment.

Text Change: A clear description of figure 4 is now included in the manuscript.

There was a significant reduction (p<0.00005) in the viability at 24 and 48 h when compared to the control at all concentrations of aANGP (Figure 4c). aANGP-MCs showed significant reduction in the viability at 24 h for concentrations of 4 mg/ml (76.9±0.25%) and 2 mg/ml (80.25±0.19%) as shown in Figure 4d. However, TC50 of aANGP peptide was obtained at 2 mg/ml (55.28±0.3%) in comparison to 4 mg/ml (58.7±0.67%) of aANGP-MCs, indicating that the efficiency of peptide interaction with cells in micellar form (Figure 4c&d). In addition, the results also showed that in the absence of starvation (Figure 4a) and starvation +VEGF treatment (Figure 4b) the peptide did not evoke any reduction in cell growth, demonstrating the specificity of the peptide to integrin avb3 upregulated in the presence of bFGF.

Text Change in methods section 2.8: The same experiment was conducted using non-starved + bFGF treated cells and starved +VEGF (20 ng/ml) treated cells.

Comment 8: Section 3.3 is related to figure 4 but there is no mention in the results to the single panels of the figure. Please add this in the text. Line 397 please check what is (b&d).

Answer: We apologize for not indicating figure 4 in the text. The section 3.3 is now modified.

Text Change: There was a significant reduction (p<0.00005) in the viability at 24 and 48 h when compared to the control at all concentrations of aANGP (Figure 4c). aANGP-MCs showed significant reduction in the viability at 24 h for concentrations of 4 mg/ml (76.9±0.25%) and 2 mg/ml (80.25±0.19%) as shown in Figure 4d. However, TC50 of aANGP peptide was obtained at 2 mg/ml (55.28±0.3%) in comparison to 4 mg/ml (58.7±0.67%) of aANGP-MCs, indicating that the efficiency of peptide interaction with cells in micellar form (Figure 4c&d). In addition, the results also showed that in the absence of starvation (Figure 4a) and starvation +VEGF treatment (Figure 4b) the peptide did not evoke any reduction in cell growth, demonstrating the specificity of the peptide to integrin avb3 upregulated in the presence of bFGF.

Comment 9: The first part of the discussion (line 419-431) is a repetition of introduction. Please remove it and stress what is the novelty of the paper.

Answer: Thank you for the comment. We modified the discussion section by deleting this repetitive information.

Comment 10: An extensive check of English grammar is necessary throughout the manuscript.

Answer: Thank you for the comment. We have done a thorough check for the spelling and grammar.

Round 2

Reviewer 2 Report

All the referee's comments have been discussed and satisfied. The mansucript has been improved in clarity and soundness or findings.